microbiology, ecology, health and disease and epidemiology

growth rate, conjugation rate, antimicrobial resistance plasmid, spent medium

**Authors for correspondence:**
Sarah J. N. Duxbury
e-mail: sjn.duxbury@gmail.com
J. Arjan G. M. de Visser
e-mail: arjan.devisser@wur.nl

†Present address: School of Life Sciences, University of Warwick, Coventry CV4 7AL, UK.

# Chicken gut microbiome members limit the spread of an antimicrobial resistance plasmid in *Escherichia coli*

Sarah J. N. Duxbury[1,†], Jesse B. Alderliesten[2], Mark P. Zwart[3], Arjan Stegeman[2], Egil A. J. Fischer[2] and J. Arjan G. M. de Visser[1]

[1]Laboratory of Genetics, Wageningen University, Wageningen, The Netherlands
[2]Department of Population Health Sciences, Faculty of Veterinary Medicine, Utrecht University, Utrecht, The Netherlands
[3]Department of Microbial Ecology, The Netherlands Institute of Ecology (NIOO-KNAW), Wageningen, The Netherlands

SJND, 0000-0003-1514-8617; JBA, 0000-0003-1132-4310; MPZ, 0000-0003-4361-7636;
AS, 0000-0003-4361-3846; EAJF, 0000-0002-0599-701X; JAGMdV, 0000-0002-5098-6686

Plasmid-mediated antimicrobial resistance is a major contributor to the spread of resistance genes within bacterial communities. Successful plasmid spread depends upon a balance between plasmid fitness effects on the host and rates of horizontal transmission. While these key parameters are readily quantified *in vitro*, the influence of interactions with other microbiome members is largely unknown. Here, we investigated the influence of three genera of lactic acid bacteria (LAB) derived from the chicken gastrointestinal microbiome on the spread of an epidemic narrow-range ESBL resistance plasmid, IncI1 carrying $bla_{CTX-M-1}$, in mixed cultures of isogenic *Escherichia coli* strains. Secreted products of LAB decreased *E. coli* growth rates in a genus-specific manner but did not affect plasmid transfer rates. Importantly, we quantified plasmid transfer rates by controlling for density-dependent mating opportunities. Parametrization of a mathematical model with our *in vitro* estimates illustrated that small fitness costs of plasmid carriage may tip the balance towards plasmid loss under growth conditions in the gastrointestinal tract. This work shows that microbial interactions can influence plasmid success and provides an experimental-theoretical framework for further study of plasmid transfer in a microbiome context.

## 1. Introduction

Plasmids are primary drivers of the spread and persistence of antimicrobial resistance genes in human and animal microbiomes [1–4]. Resistance genes with functions in antibiotic inactivation occur particularly often on plasmids in clinical settings [5,6], and are commonly transmitted horizontally between bacterial strains and species via cell–cell contacts in a process called conjugation [7,8]. Given that resistance genes can transfer from commensal bacteria to pathogens [2], a quantitative understanding of plasmid transmission is needed to address unwanted resistance evolution [9].

Plasmid spread depends on both the fitness effects of plasmid carriage and the rate of plasmid transfer [4,10–13]. Initially, plasmids typically impose a metabolic burden on their bacterial hosts in the absence of positive selection for plasmid-encoded genes [14], but plasmid maintenance is enhanced by addiction systems [1] and compensatory adaptations that reduce or ameliorate plasmid fitness costs [15–17]. Plasmid conjugation itself is also energetically costly [14], so when conjugation opportunities are limited, adaptation may lead to reduced conjugation rates [18–20]. In the absence of positive plasmid selection, horizontal transmission can maintain costly plasmids [4,10,12], particularly when

transfer is possible to a broad host range [21,22]. Since plasmid transfer rates are limited by mating opportunities of donor and recipient bacteria, and hence are affected by bacterial densities and growth rates, it is crucial to infer conjugation rates using population-dynamic models that can control for cell contact opportunities [23].

Plasmid-mediated antimicrobial resistance is expected to play an important role in resistance spread in dense microbial consortia, such as the gastrointestinal microbiome of humans [2,24] and animals [25,26]. Yet the effect of interactions with microbiome members on the spread of antimicrobial resistance plasmids is largely unknown. Rather, 'plasmid permissiveness' has been used to describe the diversity of bacterial taxa or strains to which plasmids can horizontally transfer [21,24,25,27]. Microbes engage in diverse social interactions, both cooperative and competitive [28–30], which are in part mediated through metabolic interactions [31], affecting the fitness and conjugation rate of focal species [32–35]. A better understanding of the influence of microbiome members on plasmid transfer is of particular relevance for predicting the prevalence of persistent plasmids such as multi-resistant narrow-host-range plasmids in *Escherichia coli* [24] and broad-host-range plasmids across bacterial taxa [27].

We tested the influence of microbiome taxa on the spread of an epidemic antimicrobial resistance plasmid of high importance. Narrow-host-range IncI1 plasmids carrying the $bla_{CTX-M-1}$ gene which encodes an extended-spectrum β-lactamase (ESBL) [36] are primarily detected in *E. coli* and *Salmonella* species [37]. This resistance gene–plasmid combination has been reported as the most abundant in Dutch broiler chickens [38] and in other livestock animals (including dairy cattle and slaughter pigs) in a recent 10-year surveillance study [5], despite a steady decrease in the usage of β-lactam antibiotics in The Netherlands over the past 5–10 years [39]. To test the role of microbiome secretions on the spread of this plasmid in *E. coli* populations, we measured growth rates and plasmid transfer rates in spent media from lactic acid bacteria (LAB) isolated from the chicken caecal microbiome.

The chicken caecum contains the highest microbial density and diversity of the intestinal tract [40,41], and its community composition is influenced by a range of host and environmental factors [42,43]. The microbiome plays an important role in metabolic fermentation and pathogen exclusion [42], with LAB described to significantly benefit gut health, leading to their development as probiotics [44,45]. LAB in broiler chickens belong to the families *Enterococcaceae* and *Lactobacillaceae*, of which *Lactobacillaceae* are the third most abundant family in the caecum, constituting approximately 10% relative abundance [41]. Other abundant families in the adult chicken caecum consist of the *Ruminococcaceae* (35% relative abundance), *Lachnospiraceae* (35% relative abundance) and *Bifidobacteriaceae* (7% relative abundance). All other families each constitute less than 2% of the caecal microbiome, including the *Enterococcaceae* that occur below 0.1%.

We found that spent media derived from LAB reduced growth rates of plasmid-carrying and plasmid-free *E. coli* strains, but did not significantly affect conjugation rates, when controlling for cell density variation. Using a continuous-flow mathematical model, we showed that minor changes in plasmid fitness costs could nevertheless shift the balance from plasmid fixation to plasmid loss under culture conditions mimicking growth in the chicken caecum.

## 2. Methods

### (a) *Escherichia coli* strains

We obtained an IncI1 plasmid carrying a $bla_{CTX-M-1}$ resistance gene from a natural chicken *E. coli* isolate (ESBL-375) (kind gift from Michael Brouwer, Wageningen Bioveterinary Research, Lelystad, The Netherlands) and transferred the plasmid via conjugation to a laboratory strain of *E. coli* (MG1655) (strain DA28200 described in [46]). This strain was used as the donor (D) in experimental conjugation assays and carried a chromosomal nalidixic acid resistance marker (NAL), isolated as a spontaneous mutant. The recipient strain (R) for conjugation assays was a differentially labelled strain of the MG1655 background, carrying chromosomal chloramphenicol resistance (CAM) (strain DA28200 described in [46] with an integrated *cat* gene cassette; GenBank accession number KM018300). IncI1 plasmid carriage was confirmed in strains ESBL-375 and D using the PBRT (PCR-Based Replicon Typing) 2.0 kit (Diatheva, Cartoceto, Italy). Transconjugants formed during conjugation assays were derived from the recipient strain with CAM resistance background and carried the IncI1 plasmid. Strains were cultured in Viande-Levure (VL) medium mimicking the chicken intestinal environment [47,48] at pH 5.9 buffered with MES (final concentration 0.1 M). Strains were preserved at −80°C in LB medium ($10 \, g \, l^{-1}$ (bacto) tryptone, $5 \, g \, l^{-1}$ yeast extract, $10 \, g \, l^{-1}$ NaCl) mixed with glycerol (final concentration 20% v/v).

### (b) Chicken caecal microbiota samples and supernatant spent medium preparation

Caecal samples were collected from six 37-day-old Ross 308 broiler chickens across three different houses from the same conventional broiler farm. Samples were denoted as communities 1–6 with grouping by houses as follows: house 1—communities 1 and 6; house 2—communities 3 and 4; and house 3—communities 2 and 5. All chickens were healthy and free of *Salmonella* species infection and received no antibiotic treatments, with the last feed being free of coccidiostats for 11 days. Chickens were slaughtered before transport to Utrecht University for caecal material collection. Caecal material was transported to Wageningen University for experimental assays and was mixed with PBS and glycerol (final concentration 20% v/v) before freezing at −80°C, under aerobic conditions. A single aliquot per sample was revived and sub-cultured in VL medium in a two-step dilution and culturing process. This involved 1 : 10 dilution in 800 µl volume in a capped 1.5 ml Eppendorf tube with growth at 41°C and 750 r.p.m., followed by 1 : 100 dilution in 10 ml volume in a capped 15-ml falcon tube with growth at 41°C and 250 r.p.m. An aliquot of each sample was frozen, and the two-step dilution and culturing process was repeated following revival, achieving dense, viable communities. Each sample was split into triplicate aliquots and frozen with glycerol (20% v/v), for use as three biological replicates for supernatant collection, to be used in the *E. coli* growth and conjugation assays. Replicates were revived via 1 : 10 dilution in VL medium and grown at 41°C and 250 rpm for 48 h to ensure replicate viability, with culturing under aerobic conditions. Aliquots of each replicate were again frozen at −80°C.

To prepare supernatants/spent media, a laboratory-grown frozen aliquot of each caecal sample was revived using a 1 : 10 dilution in VL medium (8 ml culture volume). Cultures were grown at 41°C and 250 r.p.m. for 48 h under aerobic conditions and a single aliquot of each was preserved in 20% v/v glycerol for later DNA extraction. Cultures were centrifuged (10 min at 4000 r.p.m.), supernatant was collected and each sample was filter-sterilized through a 0.22 µm filter unit. In pilot tests, supernatant media showed dose-dependent growth inhibition of the natural strain ESBL-375 and strain R (electronic supplementary

*Proc. R. Soc. B* **288**: 20212027

material, figure S1), following a dilution range of supernatant in fully nutrient-replenished media (see electronic supplementary material, Methods). For all experimental assays, supernatants were mixed with VL medium (with 1.25× all nutrients), such that supernatant composed 20% of the final culture volume (causing moderate growth inhibition) and nutrients were replenished to 1–1.2×, depending on depletion level in the supernatant. By diluting spent media products, we mimicked dilution of these bacterial taxa within the caecum. Two medium controls were included: 1.0× VL and 1.2× VL.

## (c) 16S rRNA gene characterization of laboratory-cultured caecal samples

A single frozen 1 ml aliquot per biological replicate of each community was revived and cultured in 13 ml of VL medium for up to 48 h at 41°C and 250 r.p.m. DNA was extracted using the Gentra Puregene Yeast/Bact DNA purification kit (Qiagen cat no./ID: 158567) and quantified on a Nanodrop spectrophotometer and Qubit fluorometer. DNA was sent to LGC Genomics GmbH (Berlin, Germany) for library preparation and Illumina MiSeq sequencing of the V3/V4 hypervariable region (341F-785R) of the 16S rRNA gene [49]. Primer clipped sequencing reads were processed using the DADA2 pipeline (v. 3.10) [50] in R v. 3.6.1 [51] (see electronic supplementary material for data analysis parameters). Analyses revealed enrichment of three genera of LAB (figure 1a).

## (d) *Escherichia coli* strain monoculture growth assays

Monocultures of strains D (donor) and R (recipient) were growth-profiled alongside strain T (transconjugant). Growth rates were used as input values for conjugation rate estimation (see the below section). In addition, monocultures of natural isolate ESBL-375 were growth-profiled. Strain T was isolated from a prior conjugation assay involving the transfer of IncI1 from natural strain ESBL-375 to strain R in 1.0x_VL medium. IncI1 carriage was confirmed via PBRT and chromosomal background (CAM resistance marker) was validated by PCR amplification of the fluorescence-resistance gene cassette using primers cat J23101F/R [46].

Each strain was revived from frozen stock by streaking a 5 µl aliquot on VL agar and growing overnight at 37°C then an overnight culture was prepared by inoculating a single colony into 3 ml VL medium. Cultures were then diluted by 1 : 100 in VL medium and grown into exponential phase at 37°C and 250 rpm until $OD_{600}$ (optical density at 600 nm) reached 0.4. Each culture was diluted to approximately $2 \times 10^6$ CFU ml$^{-1}$ in 1× VL and 1.2× VL media (approx. 100-fold dilution), then further diluted approximately twofold and aliquoted into eight different culture media (two control and six supernatant media) across eight rows of a 96-well microtiter plate (well volume 200 µl), in duplicate. A set of culture-free control media/supernatant wells were also added across eight wells in duplicate. Fifty microlitres of sterile mineral oil were added to the surface of each culture to reduce condensation on the plate lid and evaporation. The microtiter plate was incubated in a Victor3 plate reader (Perkin and Elmer, MA, USA) at 37°C under static conditions, with $OD_{600}$ readings taken every 6 min up to 24 h. The assay was repeated with three biological replicates using independent cultures and supernatant collection.

## (e) Conjugation assays and conjugation rate estimation

We performed conjugation assays between strains D and R, consisting of an initial mixed strain ratio of 50 : 50. Initial strain densities were approximately $10^6$ CFU ml$^{-1}$ and aliquots were added to the eight different culture media (described above) and incubated in separate wells of the same 96-well plate (per

biological replicate assay) as described for monoculture growth (section above). Appropriate serial dilutions from each mixed culture were performed at $t = 0$ and 4 h and plated on antibiotic-containing VL agar. Strain D was selected on CTX + NAL (cefotaxime, 1 mg l$^{-1}$ + nalidixic acid, 20 µg ml$^{-1}$). Strain R was selected on CAM (chloramphenicol, 32 µg ml$^{-1}$) and transconjugants were selected on CTX + CAM. Transconjugant counts were subtracted from counts on CAM agar to deduce recipient densities. Plates were incubated at 37°C for up to 24 h.

To estimate the endpoint conjugation rate, we used the population-dynamic 'approximate extended Simonsen' method (equation (2.1)) [23], which is an extension of the endpoint method [52], allowing for differences in growth rates ($\psi$) between donor, recipient and transconjugant strains. We calculated the conjugation rate ($\gamma$) for strain D with input of D, R and T monoculture growth rates ($\psi_D$, $\psi_R$ and $\psi_T$). D, R and T represent the endpoint cell densities ($t = 4$ h), for the donor and recipient strains and transconjugants formed during the experiment. $D_0$ and $R_0$ represent the initial cell densities from mixed cultures,

$$\gamma = (\psi_D + \psi_R - \psi_T) \frac{T}{DR - D_0 R_0 e^{\psi_T t}}. \tag{2.1}$$

Conjugation rates were log$_{10}$-transformed for statistical analyses.

## (f) Model of within-population dynamics of plasmid-bearing bacteria

We defined a simple mathematical model of the dynamics of a bacterial population containing cells with or without a plasmid to show the qualitative outcomes for population dynamics of plasmid-bearing bacteria in a continuous-flow system, given the *in vitro* estimated growth and conjugation rate parameters. We consider a situation of invasion of the plasmid into a population through an introduction by an otherwise isogenic strain. The ecological model was used to determine the parameter space for plasmid loss, coexistence and fixation, assuming no adaptation of the plasmid or bacterial host. The border of the parameter regions was determined by the population density, because we assumed a linear dependence of conjugation opportunities on population density ($N$ term in equation (2.2)) [52]. The model assumed a bacterial population with a constant size $N$, but with a continuous turn-over of cells ($m$).

The change in fraction of plasmid-carrying bacteria $p$ at time $t$ is:

$$\frac{\mathrm{d}p}{\mathrm{d}t} = m \cdot \frac{\Delta\psi \cdot p \cdot (1 - p)}{\psi + \Delta\psi \ p} + \gamma \cdot N \cdot p \cdot (1 - p), \tag{2.2}$$

where $\psi$ is equivalent to $\psi_R$ (equation (2.1)), and $\Delta\psi = \psi_D - \psi_R$. $\Delta\psi$ was calculated per replicate. $\gamma$ is the conjugation rate from equation (2.1). The fraction of plasmid-carrying bacteria ($p$) is at equilibrium for plasmid loss ($p = 0$), at fixation of the plasmid ($p = 1$) and for intermediate values, when $p = -(\psi/\Delta\psi) - (m/(\gamma N))$.

The mathematical model was applied to three population densities, $10^6$, $10^8$ and $10^{10}$ cells ml$^{-1}$, and a turn-over rate of $m$ of 0.1 cell ml$^{-1}$ h$^{-1}$. To compare the parameter estimates between all control and supernatant media, the growth rates and conjugation rates were scaled to $\psi_R$. See electronic supplementary material for further details on derivation.

## (g) Statistical analyses and computations

$OD_{600}$ data were blank corrected with the minimum reading from all medium/supernatant culture-free control wells, per 96-well assay plate. The maximum growth rates were calculated similar to the methods of [53], identifying the maximum gradient of natural logarithm-transformed OD values in sliding 1.5 h time windows in the exponential phase (data region between 102 and 228 min), using Python version 3.6.3 [54]. All further statistical

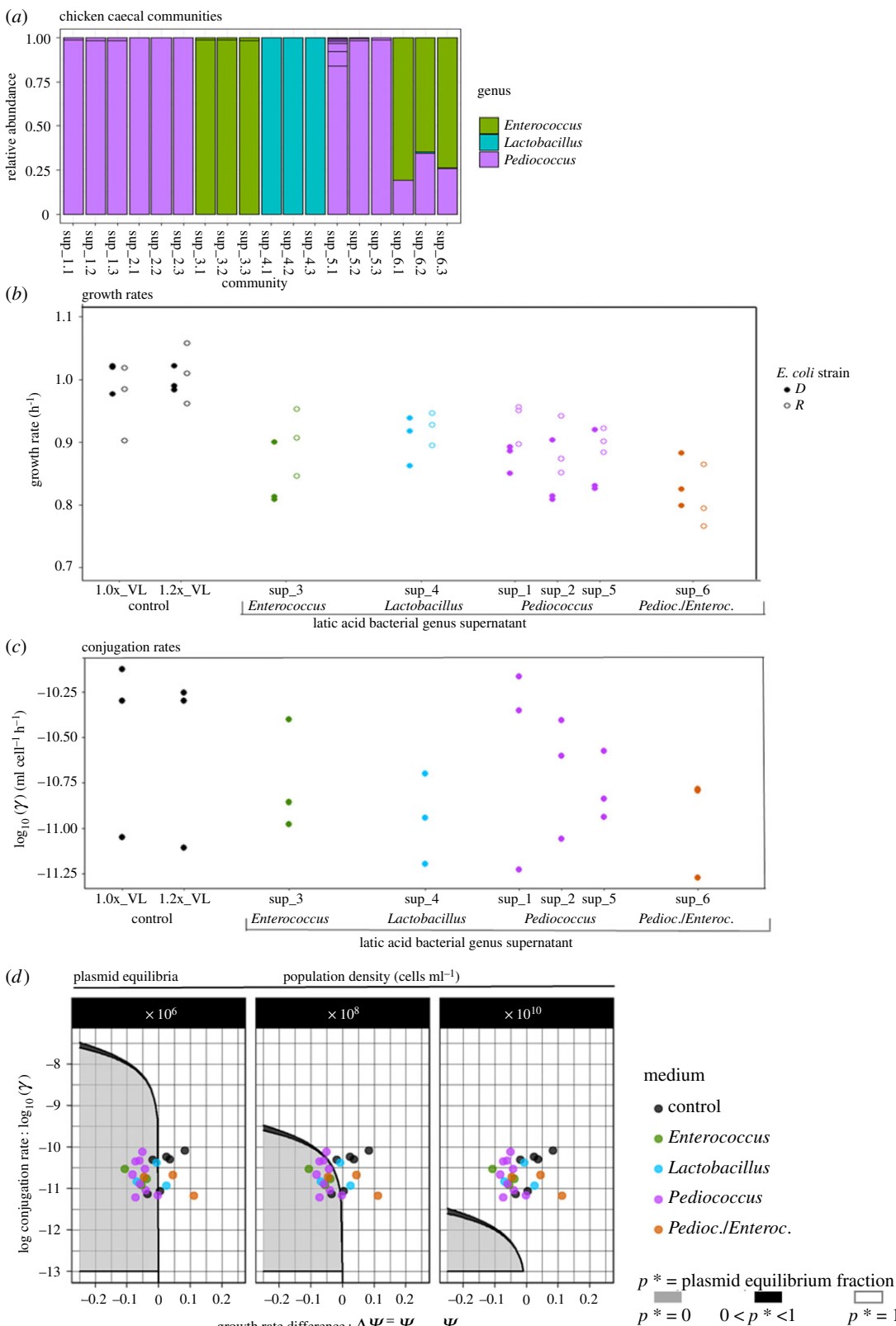

**Figure 1.** Influence of spent media from chicken microbiome-derived lactic acid bacterial genera on *E. coli* strain growth rates, plasmid conjugation rates and modelled within-population plasmid equilibria. (*a*) Spent media were collected from aerobically cultured lactic acid bacterial genera from the chicken caecum that were taxonomically identified by 16S rRNA gene sequencing. Horizontal lines on bar plots indicate different amplicon sequence variants and are plotted as relative taxonomic abundances. Triplicate replicates for six caecal samples are shown. Data points in (*b*–*d*) are three biological replicates, each averaged across two technical replicates, grouped by lactic acid bacterial genera from which supernatant was derived. (*b*) Growth rates of donor (*D*) and recipient (*R*) strains in the MG1655 background across control and supernatant-supplemented media. (*c*) Conjugation rates of IncI1 plasmid transfer between strains D and R. (*d*) Implications of lactic acid bacterial genera secreted products for expected equilibrium frequencies of plasmid IncI1 using a continuous-flow model. Shown are three possible equilibria as a function of the conjugation coefficient ($\gamma$) (presented in (*c*)) and growth rate difference between strains D and R (rates presented in (*b*)), scaled to the growth rate of the recipient strain per medium ($\Delta\psi$). Modelled population dynamics were parameterized with scaled conjugation and growth rates ($\psi_R = 1$) and an outflow rate (*m*) = 0.1, for three different population densities ranging over four $\log_{10}$ steps. (Online version in colour.)

analyses, including calculation of conjugation rates, were performed with R v. 3.6.1 [51].

For *E. coli* growth rate data in the MG1655 background, we used a two-way ANOVA to test the effects of 'strain' (donor or recipient) and 'medium' (eight groups: controls: 1.0x_VL, 1.2x_VL and treatments: Sup_1, Sup_2, Sup_3, Sup_4, Sup_5, Sup_6). For *post hoc* comparisons of each of the six treatment groups with each of the two control groups, Dunnett's tests were applied using package DescTools (v. 0.99.41) [55], per control group and strain. For conjugation rate data, we used a one-way ANOVA to test the effect of 'medium'. Test assumptions on residual distribution, normality and homoscedasticity were met.

## 3. Results

### (a) Caecal microbiota composition after *in vitro* culture

Caecal microbiota samples isolated from six broiler chickens were cultured *in vitro* under aerobic conditions. Bacterial characterization revealed enrichment of three genera of LAB (figure 1*a*). Samples 1–5 consisted of single-genera cultures: samples 1, 2 and 5 contained *Pediococcus*, in sample 3, *Enterococcus* was present and in sample 4, *Lactobacillus*. Sample 6 was enriched for two genera: approximately 75% *Enterococcus* sequences and 25% *Pediococcus* sequences. Nutrient-replenished filtered spent media from the laboratory-cultured samples were used for *E. coli* growth and conjugation assays described below.

### (b) Metabolic products of lactic acid bacteria negatively affect *Escherichia coli* donor and recipient growth rates

We investigated the influence of secreted metabolic products in the spent media produced from the three chicken-derived genera of LAB, on the growth rate of two isogenic *E. coli* strains, one carrying a conjugative IncI1-ESBL plasmid (donor, D) and one without the plasmid (recipient, R). *Escherichia coli* growth rates significantly varied across the eight medium types (two control and six supernatant media) (two-way ANOVA: $F_{7,39} = 12.31$, $p = 3.46 \times 10^{-8}$; figure 1*b*). Growth rates were on average 2.3% lower for the donor relative to the recipient strain, but this difference was not significant (two-way ANOVA: $F_{1,39} = 2.98$, $p = 0.093$).

For the donor strain, supernatant treatments significantly reduced growth rates relative to either control group (1.0x_VL or 1.2x_VL) (Dunnett's test: $p < 0.05$) for all supernatants apart from Sup_4 (*Lactobacilllus*) (electronic supplementary material, table S1). Relative to the 1.0x_VL control group, growth rate reductions ranged from 12.9% in Sup_1 (*Pediococcus*) to 16.9% in Sup_6 (*Pediococcus/Enterococcus*). Growth rate reductions in supernatant treatment groups relative to the 1.2x_VL control were highly similar (electronic supplementary material, table S1).

For the recipient strain, only Sup_6 (*Pediococcus/Enterococcus*) (electronic supplementary material, table S1) significantly reduced the growth rate relative to both control groups (Dunnett's test: $p < 0.05$). Growth rates in Sup_6 were reduced by 16.5% relative to 1.0x_VL and by 19.9% relative to 1.2x_VL. Relative to 1.2x_VL, significant growth rate reductions of 10.6–12.0% were also found in Sup_2 and 5 (*Pediococcus*) and Sup_3 (*Enterococcus*).

We also tested growth across the control and supernatant media for the natural ESBL strain isolated from a chicken source (ESBL-375), from which the IncI1 plasmid was obtained. Growth rates significantly varied across supernatant media in comparison with the 1.2x_VL control group (one-way ANOVA: $F_{6,14} = 3.08$, $p = 0.039$), but not in comparison with the 1.0x_VL control group (one-way ANOVA: $F_{6,14} = 1.19$, $p = 0.37$) (electronic supplementary material, figure S2). Average growth rates did not significantly differ between 1.0x_VL and 1.2x_VL (two-sample *t*-test: $t_4 = 0.892$, $p = 0.423$). In comparison with 1.2x_VL, growth rates were significantly reduced in supernatants 2, 5 (*Pediococcus*) and 6 (Dunnett's *post hoc* tests: $p < 0.05$) with the greatest reduction of 15.6% in Sup_6 (*Pediococcus/Enterococcus*) (electronic supplementary material, table S2).

### (c) Metabolic products of lactic acid bacteria do not significantly affect IncI1 plasmid transfer rates

We then estimated the rate of IncI1 plasmid transfer from strain D to strain R in the absence and presence of the same spent media, using the approximate extended Simonsen method to control for density-dependent variation in mating opportunities [23]. Conjugation rates did not significantly differ across medium types (two control and six supernatants) (one-way ANOVA: $F_{7,16} = 0.611$, $p = 0.739$) (figure 1*c*). Note that for supernatants 2–6, the mean conjugation rates were lower than both control groups, ranging from a 1.6–1.8-fold reduction in Sup_2 (*Pediococcus*) to a 3.0–3.5-fold reduction in Sup_5 (*Pediococcus*) and Sup_6 (*Pediococcus/Enterococcus*); however, these reductions were non-significant in pairwise Dunnett *post hoc* comparisons (electronic supplementary material, table S3).

### (d) Secreted products of lactic acid bacteria increase chance of plasmid loss under continuous-flow conditions

Next, we investigated the implications of the observed effects from microbiome members on growth rates of D and R for the possible spread of the IncI1 ESBL plasmid. Our analysis shows that there is not a statistically significant effect of microbiome members on the conjugation rate of the IncI1 plasmid from strain D. However, to consider the respective importance of growth and conjugation rates for plasmid loss, we considered the possibility that the variation in conjugation rates between supernatants represents genuine, but small differences associated with different microbiome members. We implemented a mathematical model of continuous culture conditions mimicking the chicken gastrointestinal tract. Using our *in vitro* estimates for the growth rates of strains D and R differing in IncI1 plasmid carriage (figure 1*b*) and the conjugation rates of plasmid transfer from strain D to strain R (figure 1*c*) to parametrize the model, we determined the equilibrium frequency of plasmid-carrying bacteria ($p$) for a continuous-flow system (figure 1*d*).

The model predicts that at densities of $10^8$ cells ml$^{-1}$ (relevant for the chicken caecum [56]) and lower, the three lactic acid bacterial genera increase the chance of plasmid loss compared with control media, as the data points shift towards the plasmid loss region (figure 1*d*). This shift is due mainly to increased costs of plasmid carriage in spent media shown

by greater reductions in growth rates of strain D than strain R. The model shows that the magnitude of effects of spent media on conjugation rates is too small to have a meaningful impact on plasmid loss. When cell densities are higher (e.g. $10^{10}$ cells ml$^{-1}$), the plasmid is expected to become established irrespective of plasmid costs and the range of conjugation rate values shown in our data (figure 1$d$); however, such population densities are higher than expected for *E. coli* in the chicken caecum.

## 4. Discussion

Our study makes three main contributions. First, we show that secreted products from lactic acid bacterial genera derived from the chicken gastrointestinal microbiome substantially reduce *E. coli* strain growth rates, while not causing reductions in conjugation rates. Second, using a mathematical model of continuous-flow bacterial population dynamics parametrized with our *in vitro* estimates, we predict that the secreted products would increase the chance of plasmid loss under conditions mimicking growth in the gastrointestinal tract. These effects are driven by the changes in growth rate in the parameter space that is biologically relevant. Finally, we demonstrate the utility of using spent media derived from microbiome taxa (see also [30] and [57]) to investigate effects of microbial interactions on key determinants of plasmid spread in focal *E. coli* populations, extending quantitative *in vitro* conjugation assays to more realistic conditions.

We observed a negative effect of LAB from the order *Lactobacilliales* on the growth rate of *E. coli* strains, as observed previously *in vitro* [44,58]. LAB can inhibit *E. coli* strains via bacteriocin production and resource competition, while adhered to intestinal epithelial cells [44]. Likewise, probiotic intestinal bacteria, including *Enterococcus* and *Lactobacillus*, can successfully reduce *in vivo* invasion, excretion and between-animal transmission of ESBL *E. coli* in broiler chickens [45,59]. As the order *Lactobacilliales* constitutes up to 25% of the chicken caecal microbiome [60,61], understanding its role in mediating within-microbiome plasmid spread remains an important challenge. We observed reductions in *E. coli* growth rates for our donor and recipient strains in the laboratory strain MG1655 background. For the natural ESBL strain isolated from a chicken source from which the IncI1 plasmid was obtained, we also observed growth rate reductions in supernatant media; however trends relative to controls were less clear. We recommend that future studies explore the influence of microbial secretions on growth rates of a larger range of natural strains.

We did not observe a significant effect on conjugation rates from spent media. We did observe variability over an order of magnitude between our replicate data points (as also reported for natural isolates [62]); therefore, we recommend that conjugation assays are performed with increased replication to improve the reliability of estimates. By quantifying endpoint conjugation *rates* rather than *frequencies*, we were able to control for differences in mating opportunities (due to differences in cell densities) [23], so that true kinetics of plasmid transfer could be compared across control and supernatant conditions. This presents a key methodological advance, as separating growth and transfer dynamics poses a challenge in comparing plasmid transmission rates across environmental conditions [3,63].

Plasmid success is influenced by a balance between growth rate effects of plasmid carriage, conjugation rate and population density. *Escherichia coli* has been found at densities of $10^6$–$10^8$ cells ml$^{-1}$ in the caecum of chickens [56,64]. At these population densities, we found by modelling conjugation under continuous-flow conditions, that fixation of the plasmid was the likely outcome in control media due to the minimal growth rate costs (as we also reported in batch culture [62]), whereas in the presence of secreted products of LAB, plasmid fate shifted towards loss. This shift was due mainly to a slightly increased fitness cost of IncI1 plasmid carriage ($\Delta\psi$ values), with little influence from spent media effects on conjugation rates (figure 1$d$). This may support recently reported epidemiological decreases of ESBL resistance (predominantly due to $bla_{CTX-M-1}$ carriage on IncI1 plasmids [5]) within Dutch chicken populations in association with a ban on the usage of third-generation cephalosporins [65], thereby reducing postive selection pressure for persistence of these plasmids. Our finding, however, contrasts with that in another study [4], predicting the maintenance of ESBL-plasmids (via conjugation) with greater fitness costs in the absence of antibiotics at a higher population density of $10^9$ CFU ml$^{-1}$, which confirms an important role of cell density [52]. Conjugation rates are probably lower for transfer between different bacterial strains [66] or species [63] than for isogenic strains of *E. coli*, thus conditions for plasmid spread across bacterial species are expected to depend more critically on the balance between plasmid costs and cell densities.

We present a novel approach to investigate the role of microbial interactions on plasmid spread in a focal bacterial population, using *in vitro* assays in spent media from relevant bacterial taxa. The use of spent media provides a controlled method to study the influence of microbial interactions and has been applied to measure effects on individual growth kinetics [30,57] and pairwise competition outcomes [32]. We applied this approach to investigate effects on conjugation from dominant microbiome members under aerobic conditions, but it could also be applied under anaerobic gut conditions, complementing recent studies of community effects on selection for antimicrobial resistance [33,34]. Future studies should take into account additional features of microbiomes, including spatial structuring [67,68] and influence of the host immune system [69,70], to help bridge the gap between *in vitro* and *in vivo* parameter estimates of plasmid spread.

**Ethics.** The collection of caecae was part of another study for which the animal experiment was approved by the Dutch Central Authority for Scientific Procedures on Animals and the Animal Experiments Committee (registration number AVD108002016442) and complied with all relevant legislation.

**Data accessibility.** Datasets, code and syntax are available from the Dryad Digital Repository: https://doi.org/10.5061/dryad.2jm63xsp7 [71].

Supplementary material and figures are available from Figshare [72].

**Authors' contributions.** S.J.N.D.: conceptualization, formal analysis, investigation, methodology, visualization, writing—original draft, Writing—review and editing; J.B.A.: conceptualization, formal analysis, investigation, methodology, visualization, writing—review and editing; M.P.Z.: conceptualization, funding acquisition, investigation, methodology, supervision, writing—review and editing; A.S.: conceptualization, funding acquisition, investigation, methodology, supervision, writing—review and editing; E.A.J.F.: conceptualization, formal analysis, funding acquisition, investigation, methodology, project administration, resources, supervision, validation, visualization, writing—original draft, writing—review and editing;

J.A.G.M.d.V.: conceptualization, funding acquisition, investigation, methodology, supervision, writing—original draft, writing—review and editing.

All authors gave final approval for publication and agreed to be held accountable for the work performed therein.

Competing interests. We declare we have no competing interests.

Funding. This work was funded by a ZonMW grant (grant no. 541001005) to Utrecht University and Wageningen University.

Acknowledgements. We thank Francisca Velkers (Utrecht University) for collecting the caecal samples and Jan van den Broek for advice on statistical analyses. We also thank Michael Brouwer, Ingrid Cardenas Rey and Dik Mevius (Wageningen Bioveterinary Research, Lelystad) for providing the ESBL-375 *E. coli* strain, advice on bioinformatics analyses and for stimulating discussions. We thank Philip Ruelens (Wageningen University) for sharing Python code for growth rate estimation.

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
