## [Peer Review File · Proceedings of the Royal Society B: Biological Sciences]

Review History

RSPB-2021-0909.R0 (Original submission)

Review form: Reviewer 1

Recommendation

Major revision is needed (please make suggestions in comments)

Scientific importance: Is the manuscript an original and important contribution to its field?

Acceptable

General interest: Is the paper of sufficient general interest?

Acceptable

Quality of the paper: Is the overall quality of the paper suitable?

Acceptable

Is the length of the paper justified?

Yes

Should the paper be seen by a specialist statistical reviewer?

No

Do you have any concerns about statistical analyses in this paper? If so, please specify them explicitly in your report.

No

It is a condition of publication that authors make their supporting data, code and materials available - either as supplementary material or hosted in an external repository. Please rate, if applicable, the supporting data on the following criteria.

Is it accessible?

Yes

Is it clear?

Yes

Is it adequate?

Yes

Do you have any ethical concerns with this paper?

No

Comments to the Author

In this study, the authors investigate the effect of secreted products of lactic acid bacteria from chicken gastrointestinal microbiome on the stability of a narrow-host range antibiotic resistance conjugative plasmid. They show that these products reduce the growth rate of the *E. coli* MG1655 strain, used as a model system in the study, and may also reduce conjugation rate. Using a mathematical model incorporating the data obtained from the experimental system, the authors show that these secreted products of lactic acid bacteria may generally increase chance of plasmid loss in a bacterial community. There are several aspects that I find interesting in this study such as trying to incorporate the effects of microbial interactions on the "existence conditions of plasmids", using a natural antibiotic resistance plasmid common in the chicken gut, or using natural chicken gastrointestinal samples to recover real chicken gut microbiome. However, I also have some concerns that I enumerate below:

-The choice of strain for the experiments. While the authors used real chicken gastrointestinal samples to obtain the secreted products from lactic acid bacteria, and a natural *E. coli* isolate from a chicken to obtain the plasmid, they then moved the plasmid to *E. coli* MG1655 to perform all the experiments. *E. coli* MG1655 derives from the original K-12 isolate, which was obtained from a stool sample of a diphtheria patient in Palo Alto, CA in 1922, (sequence type 10). MG1655 has a human origin. I wonder if the effects observed would have been the same if the authors have chosen an *E. coli* isolate from the chicken gut microbiota, which may be adapted to growing in the presence of the specific lactic acid bacteria recovered from the chicken gut. I understand that repeating all the experiments with a different *E. coli* isolated from the chicken gut may be a bit unrealistic, but I think using MG1655 may be an important limitation for the interpretation of the results.

-While the changes in growth rates in the presence of the secreted products are quite evident, the changes in conjugation rates are a less clear. Measuring conjugation rates is a bit noisy and, at least in our experience, it is useful to do at least 5-6 independent replicates to determine the conjugation rates.

-The culture conditions of the gut microbiota (multiple subculture steps in aerobic conditions) could bias the composition of the gut microbiota, enriching the lactic acid bacteria? If this was the case then the concentration of secreted products may be higher than those reached in the gut.

I am not an expert on mathematical modelling, so I cannot really assess the quality of their model in detail.

Review form: Reviewer 2

Recommendation

Major revision is needed (please make suggestions in comments)

Scientific importance: Is the manuscript an original and important contribution to its field?

Good

General interest: Is the paper of sufficient general interest?

Good

Quality of the paper: Is the overall quality of the paper suitable?

Good

Is the length of the paper justified?

Yes

Should the paper be seen by a specialist statistical reviewer?

No

Do you have any concerns about statistical analyses in this paper? If so, please specify them explicitly in your report.

Yes

It is a condition of publication that authors make their supporting data, code and materials available - either as supplementary material or hosted in an external repository. Please rate, if applicable, the supporting data on the following criteria.

Is it accessible?

Yes

Is it clear?

Yes

Is it adequate?

Yes

Do you have any ethical concerns with this paper?

No

Comments to the Author

In this manuscript by Duxbury et al., the authors investigate the influence of different microbiome taxa on the rate of transmission of plasmid-mediated antimicrobial resistance, by testing plasmid conjugation and fitness effects in spent media from different chicken caecal microbiota. They suggest that metabolic byproducts from resident lactic acid bacteria increase plasmid fitness costs and reduce transfer rates, and use these data to parameterise a model that demonstrates that even minor changes in fitness and transmission rates are capable of leading to plasmid loss in media mimicking the environment of the chicken caecum. Overall the subject and scope of the study is appropriate for the journal considering the global significance of the dissemination of antimicrobial resistance from and between livestock. The manuscript is well-written and caveats appropriately justified. I liked that the authors include parameterised modelling simulations to explain how their findings may have biological impact.

My major concern is that key conclusions of the study rest on some subtle effects from linear mixed-effects models ($0.01 < p < 0.05$). Certainly looking at the figures (1b and 1c) does not reveal

any striking effects. I am therefore concerned that the findings emerge from the way that the data are analysed. Specifically:

- The authors pool different communities that were overrepresented in the same genus (*Pediococcus*) and analyse these all together. The authors also pool the control samples (which contain different media concentrations) and analyse these all together. This seems rather arbitrary and results in an unbalanced design. It may also increase the power for the *Pediococcus* comparison, potentially explaining why this genus (but not the others) had a significant effect on the growth rates of plasmid-carriers. Does the key finding hold, if these are analysed separately?

- The authors ran their experiments three times (in different 96-well plates) and include this 'batch effect' as a random effect on intercept in their model. However, with only three levels of random effect, it is difficult to estimate the among-block variance (some suggest that random effects require >5 levels, for example Ben Bolker <http://bbolker.github.io/mixedmodels-misc/glmmFAQ.html#should-i-treat-factor-xxx-as-fixed-or-random>). Is there a danger of model overfitting?

The authors should explain how their results are robust to alternative analysis approaches, or explain clearly how their specific approach was planned and justified. Ideally the key strains would also be isolated from the mixed cultures and their effects on plasmid dynamics investigated, but this would be a considerable amount of work.

Minor suggestions/comments:

- A clearer justification of why LAB were selected for this study would have been helpful. Although their relative abundance in the chicken caecum environment are stated, putting this into broader context would be useful, such as discussion of other dominant members and their abundances, along with brief information on what is currently known about the microbiome of chicken gut.

- Why were these experiments done under aerobic conditions, when the majority of gut bacteria are anaerobes? Or does adding the oil to the plate in the plate reader result in an anaerobic culture?

Decision letter (RSPB-2021-0909.R0)

07-May-2021

Dear Dr Duxbury:

I am writing to inform you that your manuscript RSPB-2021-0909 entitled "Chicken gut microbiome members limit the spread of an antimicrobial resistance plasmid in *Escherichia coli*" has, in its current form, been rejected for publication in Proceedings B.

This action has been taken on the advice of referees, who have recommended that substantial revisions are necessary. With this in mind we would be happy to consider a resubmission, provided the comments of the referees are fully addressed. However please note that this is not a provisional acceptance.

The resubmission will be treated as a new manuscript. However, we will approach the same reviewers if they are available and it is deemed appropriate to do so by the Editor. Please note that resubmissions must be submitted within six months of the date of this email. In exceptional

circumstances, extensions may be possible if agreed with the Editorial Office. Manuscripts submitted after this date will be automatically rejected.

Sincerely,
Dr Sasha Dall
mailto:proceedingsb@royalsociety.org

Associate Editor
Board Member: 1
Comments to Author:

Your manuscript has been reviewed by two experts in the field. Both found the ideas motivating your research novel, the combination of experiments and models compelling, and the results potentially interesting. However, both reviewers raise some significant concerns. The major issues are: 1. the choice/relevance of bacterial strains used in your experiments; 2. the statistical analysis and interpretation of your data.

Reviewer(s)' Comments to Author:
Referee: 1

Comments to the Author(s)

In this study, the authors investigate the effect of secreted products of lactic acid bacteria from chicken gastrointestinal microbiome on the stability of a narrow-host range antibiotic resistance conjugative plasmid. They show that these products reduce the growth rate of the *E. coli* MG1655 strain, used as a model system in the study, and may also reduce conjugation rate. Using a mathematical model incorporating the data obtained from the experimental system, the authors show that these secreted products of lactic acid bacteria may generally increase chance of plasmid loss in a bacterial community. There are several aspects that I find interesting in this study such as trying to incorporate the effects of microbial interactions on the "existence conditions of plasmids", using a natural antibiotic resistance plasmid common in the chicken gut, or using natural chicken gastrointestinal samples to recover real chicken gut microbiome. However, I also have some concerns that I enumerate below:

-The choice of strain for the experiments. While the authors used real chicken gastrointestinal samples to obtain the secreted products from lactic acid bacteria, and a natural *E. coli* isolate from a chicken to obtain the plasmid, they then moved the plasmid to *E. coli* MG1655 to perform all the experiments. *E. coli* MG1655 derives from the original K-12 isolate, which was obtained from a stool sample of a diphtheria patient in Palo Alto, CA in 1922, (sequence type 10). MG1655 has a human origin. I wonder if the effects observed would have been the same if the authors have chosen an *E. coli* isolate from the chicken gut microbiota, which may be adapted to growing in the presence of the specific lactic acid bacteria recovered from the chicken gut. I understand that

repeating all the experiments with a different *E. coli* isolated from the chicken gut may be a bit unrealistic, but I think using MG1655 may be an important limitation for the interpretation of the results.

-While the changes in growth rates in the presence of the secreted products are quite evident, the changes in conjugation rates are a less clear. Measuring conjugation rates is a bit noisy and, at least in our experience, it is useful to do at least 5-6 independent replicates to determine the conjugation rates.

-The culture conditions of the gut microbiota (multiple subculture steps in aerobic conditions) could bias the composition of the gut microbiota, enriching the lactic acid bacteria? If this was the case then the concentration of secreted products may be higher than those reached in the gut.

I am not an expert on mathematical modelling, so I cannot really assess the quality of their model in detail.

Referee: 2

Comments to the Author(s)

In this manuscript by Duxbury et al., the authors investigate the influence of different microbiome taxa on the rate of transmission of plasmid-mediated antimicrobial resistance, by testing plasmid conjugation and fitness effects in spent media from different chicken caecal microbiota. They suggest that metabolic byproducts from resident lactic acid bacteria increase plasmid fitness costs and reduce transfer rates, and use these data to parameterise a model that demonstrates that even minor changes in fitness and transmission rates are capable of leading to plasmid loss in media mimicking the environment of the chicken caecum. Overall the subject and scope of the study is appropriate for the journal considering the global significance of the dissemination of antimicrobial resistance from and between livestock. The manuscript is well-written and caveats appropriately justified. I liked that the authors include parameterised modelling simulations to explain how their findings may have biological impact.

My major concern is that key conclusions of the study rest on some subtle effects from linear mixed-effects models ($0.01 < p < 0.05$). Certainly looking at the figures (1b and 1c) does not reveal any striking effects. I am therefore concerned that the findings emerge from the way that the data are analysed. Specifically:

- The authors pool different communities that were overrepresented in the same genus (*Pediococcus*) and analyse these all together. The authors also pool the control samples (which contain different media concentrations) and analyse these all together. This seems rather arbitrary and results in an unbalanced design. It may also increase the power for the *Pediococcus* comparison, potentially explaining why this genus (but not the others) had a significant effect on the growth rates of plasmid-carriers. Does the key finding hold, if these are analysed separately?
- The authors ran their experiments three times (in different 96-well plates) and include this 'batch effect' as a random effect on intercept in their model. However, with only three levels of random effect, it is difficult to estimate the among-block variance (some suggest that random effects require >5 levels, for example Ben Bolker <http://bbolker.github.io/mixedmodels-misc/glmmFAQ.html#should-i-treat-factor-xxx-as-fixed-or-random>). Is there a danger of model overfitting?

The authors should explain how their results are robust to alternative analysis approaches, or explain clearly how their specific approach was planned and justified. Ideally the key strains would also be isolated from the mixed cultures and their effects on plasmid dynamics investigated, but this would be a considerable amount of work.

Minor suggestions/comments:

- A clearer justification of why LAB were selected for this study would have been helpful. Although their relative abundance in the chicken caecum environment are stated, putting this

into broader context would be useful, such as discussion of other dominant members and their abundances, along with brief information on what is currently known about the microbiome of chicken gut.

- Why were these experiments done under aerobic conditions, when the majority of gut bacteria are anaerobes? Or does adding the oil to the plate in the plate reader result in an anaerobic culture?

Author's Response to Decision Letter for (RSPB-2021-0909.R0)

See Appendix A.

RSPB-2021-2027.R0

Review form: Reviewer 1

Recommendation

Accept as is

Scientific importance: Is the manuscript an original and important contribution to its field?

Good

General interest: Is the paper of sufficient general interest?

Good

Quality of the paper: Is the overall quality of the paper suitable?

Good

Is the length of the paper justified?

Yes

Should the paper be seen by a specialist statistical reviewer?

No

Do you have any concerns about statistical analyses in this paper? If so, please specify them explicitly in your report.

No

It is a condition of publication that authors make their supporting data, code and materials available - either as supplementary material or hosted in an external repository. Please rate, if applicable, the supporting data on the following criteria.

Is it accessible?

Yes

Is it clear?

Yes

Is it adequate?

Yes

Do you have any ethical concerns with this paper?

No

Comments to the Author

The authors have successfully addressed all my comments.

Minor comments:

-line 81, there is an extra "("

-line 86, not sure I get the meaning (is it 10%).

Review form: Reviewer 2

Recommendation

Accept as is

Scientific importance: Is the manuscript an original and important contribution to its field?

Acceptable

General interest: Is the paper of sufficient general interest?

Good

Quality of the paper: Is the overall quality of the paper suitable?

Good

Is the length of the paper justified?

Yes

Should the paper be seen by a specialist statistical reviewer?

No

Do you have any concerns about statistical analyses in this paper? If so, please specify them explicitly in your report.

No

It is a condition of publication that authors make their supporting data, code and materials available - either as supplementary material or hosted in an external repository. Please rate, if applicable, the supporting data on the following criteria.

Is it accessible?

Yes

Is it clear?

Yes

Is it adequate?

Yes

Do you have any ethical concerns with this paper?

No

Comments to the Author

In their revision, the authors have added in new data showing the effects on the original (chicken-derived) plasmid-bearing strain, and have reanalysed their results and adjusted their conclusions as a consequence. The changes to the manuscript do lessen the impact slightly, but increase the reliability and generalisability. It is a bit disappointing that (as implied by the conclusion) further

studies with increased replication are required to address some of the questions the authors set out to investigate. Nevertheless, I'm satisfied with the changes the authors have made.

Decision letter (RSPB-2021-2027.R0)

08-Oct-2021

Dear Dr Duxbury

I am pleased to inform you that your manuscript RSPB-2021-2027 entitled "Chicken gut microbiome members limit the spread of an antimicrobial resistance plasmid in *Escherichia coli*" has been accepted for publication in Proceedings B.

The referee(s) have recommended publication, but also suggest some minor revisions to your manuscript. Therefore, I invite you to respond to the referee(s)' comments and revise your manuscript. Because the schedule for publication is very tight, it is a condition of publication that you submit the revised version of your manuscript within 7 days. If you do not think you will be able to meet this date please let us know.

Online supplementary material will also carry the title and description provided during submission, so please ensure these are accurate and informative. Note that the Royal Society will not edit or typeset supplementary material and it will be hosted as provided. Please ensure that

the supplementary material includes the paper details (authors, title, journal name, article DOI). Your article DOI will be 10.1098/rspb.[paper ID in form xxxx.xxxx e.g. 10.1098/rspb.2016.0049].

[http://datadryad.org/submit?journalID=RSPB&manu=\(Document not available\)](http://datadryad.org/submit?journalID=RSPB&manu=(Document%20not%20available)) which will take you to your unique entry in the Dryad repository. If you have already submitted your data to dryad you can make any necessary revisions to your dataset by following the above link. Please see <https://royalsociety.org/journals/ethics-policies/data-sharing-mining/> for more details.

Sincerely,

Dr Sasha Dall

Associate Editor

Board Member

Comments to Author:

Thank you for revising your manuscript. Both of the original reviewers are satisfied with the changes you have made. There are a couple of minor corrections/clarifications to make.

Reviewer(s)' Comments to Author:

Referee: 1

Comments to the Author(s).

The authors have successfully addressed all my comments.

Minor comments:

-line 81, there is an extra "("

-line 86, not sure I get the meaning (is it 10%?).

Referee: 2

Comments to the Author(s).

In their revision, the authors have added in new data showing the effects on the original (chicken-derived) plasmid-bearing strain, and have reanalysed their results and adjusted their conclusions as a consequence. The changes to the manuscript do lessen the impact slightly, but increase the reliability and generalisability. It is a bit disappointing that (as implied by the conclusion) further studies with increased replication are required to address some of the questions the authors set out to investigate. Nevertheless, I'm satisfied with the changes the authors have made.

Author's Response to Decision Letter for (RSPB-2021-2027.R0)

See Appendix B.

Decision letter (RSPB-2021-2027.R1)

12-Oct-2021

Dear Dr Duxbury

I am pleased to inform you that your manuscript entitled "Chicken gut microbiome members limit the spread of an antimicrobial resistance plasmid in *Escherichia coli*" has been accepted for publication in Proceedings B.

Your article has been estimated as being 9 pages long. Our Production Office will be able to confirm the exact length at proof stage.

Data Accessibility section

Open Access

Paper charges

Sincerely,

Appendix A

Responses to reviewer comments: manuscript RSPB-2021-0909

Associate Editor

Board Member: 1

Comments to Author:

Your manuscript has been reviewed by two experts in the field. Both found the ideas motivating your research novel, the combination of experiments and models compelling, and the results potentially interesting. However, both reviewers raise some significant concerns. The major issues are: 1. the choice/relevance of bacterial strains used in your experiments; 2. the statistical analysis and interpretation of your data.

We would like to thank the reviewers and the editor for their time in reviewing this manuscript and for their helpful recommendations for improving robustness of the data presented in the manuscript. We have fully addressed all referee comments including a revision of the statistical analyses. We have also added growth rate data for the natural plasmid-carrying donor strain across control and supernatant media to support trends observed for our bacterial strains in the MG1655 background (as suggested by Reviewer 1).

Below are detailed point-by-point responses:

Reviewer(s)' Comments to Author:

Referee: 1

Comments to the Author(s)

In this study, the authors investigate the effect of secreted products of lactic acid bacteria from chicken gastrointestinal microbiome on the stability of a narrow-host range antibiotic resistance conjugative plasmid. They show that these products reduce the growth rate of the *E. coli* MG1655 strain, used as a model system in the study, and may also reduce conjugation rate. Using a mathematical model incorporating the data obtained from the experimental system, the authors show that these secreted products of lactic acid bacteria may generally increase chance of plasmid loss in a bacterial community. There are several aspects that I find interesting in this study such as trying to incorporate the effects of microbial interactions on the “existence conditions of plasmids”, using a natural antibiotic resistance plasmid common in the chicken gut, or using natural chicken gastrointestinal samples to recover real chicken gut microbiome.

Thank you for your positive feedback. Please note that we have now revised the statistical analyses (see responses to reviewer feedback below) to more robustly evaluate the influence of secreted products of lactic acid bacteria on growth rates and conjugation rates.

However, I also have some concerns that I enumerate below:

-The choice of strain for the experiments. While the authors used real chicken gastrointestinal samples to obtain the secreted products from lactic acid bacteria, and a natural *E. coli* isolate from a chicken to obtain the plasmid, they then moved the plasmid to *E. coli* MG1655 to perform all the experiments. *E. coli* MG1655 derives from the original K-12 isolate, which was obtained from a stool sample of a diphtheria patient in Palo Alto, CA in 1922, (sequence type 10). MG1655 has a human origin. I wonder if the effects observed would have been the same if the authors have chosen an *E. coli* isolate from the chicken gut microbiota, which may be adapted to growing in the presence of the specific lactic acid bacteria recovered from the chicken gut. I understand that repeating all the experiments with a different *E. coli* isolated from the chicken gut may be a bit unrealistic, but I think using MG1655 may be an important limitation for the interpretation of the results.

Thank you for your suggestion. Our aim was to present a novel method to test the effects of supernatant media on plasmid fitness effects and conjugation rates in a common host background, using the well-characterised *E. coli* strain MG1655 as a model. This approach could be applied across different *E. coli* strain backgrounds including natural strains, following differential labelling of two versions of the background strain to be used as donor and recipient strains.

In response to the reviewer's request, we have added data to the manuscript for measured growth rates of the natural chicken *E. coli* isolate (ESBL-375) carrying the IncI1 plasmid (same plasmid that was moved into MG1655) across the same control and supernatant media. Measurements were collected in three independent experiments simultaneously in the same assays and microtiter plates that were used to measure growth rates of the donor and recipient strains in the MG1655 background presented in the manuscript. We have added a sentence in the Methods section to reflect this addition (lines 174-175). Using one-way ANOVA tests, we tested the effect of 'medium' (all supernatant groups against a single control group) on growth rates. This data was not previously presented in the main manuscript, as we did not complete conjugation assays for IncI1 plasmid transfer between ESBL-375 and an IncI1 plasmid-free version of the same strain. Growth rate and conjugation rate data could therefore not be combined to determine conditions for plasmid maintenance.

For the natural ESBL strain, we observe reductions in growth rate relative to control group 1.2x_VL only, but growth rates did not significantly differ between the two control groups (1.0x_VL and 1.2x_VL). We have added the following text to the Results section on lines 326-336 and in the Supplementary material, a data figure (Supplementary Figure 2) and statistical results of pairwise post-hoc Dunnett test comparisons between control and supernatant treatment groups (Supplementary Table 2).

"We also tested growth across the control and supernatant media for the natural ESBL strain isolated from a chicken source (ESBL-375), from which the IncI1

*plasmid was obtained. Growth rates significantly varied across supernatant media in comparison with the 1.2x_VL control group (one-way ANOVA: $F_{6,14} = 3.08$, $p = 0.039$), but not in comparison with the 1.0x_VL control group (one-way ANOVA: $F_{6,14} = 1.19$, $p = 0.37$) (Supplementary Figure 2). Average growth rates did not significantly differ between 1.0x_VL and 1.2x_VL (two-sample t-test: $t(4) = 0.892$, $p = 0.423$). In comparison with 1.2x_VL, growth rates were significantly reduced in supernatants 2, 5 (*Pediococcus*) and 6 (Dunnett's post-hoc tests: $p < 0.05$) with the greatest reduction of 15.6% in Sup_6 (*Pediococcus/Enterococcus*) (Supplementary Table 2)."*

These data support trends of growth rate reduction in supernatant media seen for the donor and recipient strains in the *E. coli* MG1655 background, suggesting that supernatant effects were not specific to the MG1655 background and may indicate a general pattern across *E. coli* strains. The significance of growth rate reductions relative to both control groups is however less clear and we extend our discussion on lines 408-414 as follows:

"We observed reductions in E. coli growth rates for our donor and recipient strains in the laboratory strain MG1655 background. For the natural ESBL strain isolated from a chicken source from which the IncI1 plasmid was obtained, we also observed growth rate reductions in supernatant media however trends relative to controls were less clear. We recommend that future studies explore the influence of microbial secretions on growth rates of a larger range of natural strains."

-While the changes in growth rates in the presence of the secreted products are quite evident, the changes in conjugation rates are a less clear. Measuring conjugation rates is a bit noisy and, at least in our experience, it is useful to do at least 5-6 independent replicates to determine the conjugation rates.

Thank you for your comment. Variability in conjugation rates across several pairs of conjugating isolates has indeed been previously reported when transconjugants were measured over a short assay, with variation in rates up to two orders of magnitude when measured across four replicate data points (Dimitriu *et al.*, 2019; *Proc. R. Soc. B*, **286**: 20191110). We acknowledge that our data also show considerable variability between replicate points particularly for the conjugation rate estimates, which limits the power of our tests. To the best of our knowledge, the scatter of conjugation rate data points is due to inherent biological variability across independent mixed cultures. We agree that increasing the number of replicate data points would have improved reliability of the data, however we have no reason to doubt the validity of the three replicate data points.

Instead, we have added this caveat in the discussion (lines 417-420) to highlight the importance of increased biological replication for conjugation rate measurement:

"We did observe variability over an order of magnitude between our replicate data points (as also reported for natural isolates [64]), therefore we recommend that

conjugation assays are performed with increased replication to improve reliability of estimates.”

-The culture conditions of the gut microbiota (multiple subculture steps in aerobic conditions) could bias the composition of the gut microbiota, enriching the lactic acid bacteria? If this was the case then the concentration of secreted products may be higher than those reached in the gut.

Thank you for your comment. Yes, we agree that sub-culturing of the chicken caecal content and isolated microbiota under aerobic conditions was likely to enrich for aerobic or facultatively anaerobic organisms, lactic acid bacteria in particular. This however meant that we could study the influence of microbial interactions from this group of bacteria separately from the whole microbiota.

By diluting down the supernatant concentration (20% of final culture volume) in the microtiter plate assays for *E. coli* growth and conjugation, we mimicked dilution of products produced from these species within the microbiota, as described on lines 153-158. We acknowledge that the cultured gut microbiota and subsequent concentrations of secreted products in our experiments cannot be extrapolated to those produced in the chicken gut. This is due to multiple differences in growth conditions (anaerobic conditions, spatial structuring, antibiotic treatment) in the gut compared to the conditions used in our lab experiments. Our aim was not to provide a definitive answer as to where and when microbiome members affect plasmid success. Instead, we present a novel method and model to provide a first estimate of the influence of a subclass of microbiome members on two key parameters of plasmid spread: 1) plasmid fitness effects and 2) plasmid transfer rate. We emphasise this point for the final paragraph of the discussion on lines 450-461. This provides a basis for future studies to test plasmid success under the influence of additional conditions of relevance to *in vivo* environments.

I am not an expert on mathematical modelling, so I cannot really assess the quality of their model in detail.

Referee: 2

Comments to the Author(s)

In this manuscript by Duxbury et al., the authors investigate the influence of different microbiome taxa on the rate of transmission of plasmid-mediated antimicrobial resistance, by testing plasmid conjugation and fitness effects in spent media from different chicken caecal microbiota. They suggest that metabolic byproducts from resident lactic acid bacteria increase plasmid fitness costs and reduce transfer rates, and use these data to parameterise a model that demonstrates that even minor changes in fitness and transmission

rates are capable of leading to plasmid loss in media mimicking the environment of the chicken caecum. Overall the subject and scope of the study is appropriate for the journal considering the global significance of the dissemination of antimicrobial resistance from and between livestock. The manuscript is well-written and caveats appropriately justified. I liked that the authors include parameterised modelling simulations to explain how their findings may have biological impact.

Thank you for your positive feedback.

My major concern is that key conclusions of the study rest on some subtle effects from linear mixed-effects models ($0.01 < p < 0.05$). Certainly looking at the figures (1b and 1c) does not reveal any striking effects. I am therefore concerned that the findings emerge from the way that the data are analysed. Specifically:

- The authors pool different communities that were overrepresented in the same genus (*Pediococcus*) and analyse these all together. The authors also pool the control samples (which contain different media concentrations) and analyse these all together. This seems rather arbitrary and results in an unbalanced design. It may also increase the power for the *Pediococcus* comparison, potentially explaining why this genus (but not the others) had a significant effect on the growth rates of plasmid-carriers. Does the key finding hold, if these are analysed separately?

Thank you for your feedback and suggested improvements for the statistical data analysis to support our findings. We acknowledge that pooling of replicates creates an unbalanced data structure. We have therefore re-analysed all growth rate and conjugation rate data treating each medium group (two controls and six supernatants) separately instead of grouping by controls or by lactic acid bacterial genus. Firstly, applying ANOVA tests we find that *E. coli* growth rates significantly differ across medium groups but not between donor and recipient strains. Using Dunnett tests per strain, we compare growth rates for each of the control groups separately with each of the treatment (supernatant) groups. For the donor, growth rates were significantly reduced relative to both control groups in all individual supernatants apart from supernatant 4 (*Lactobacillus*). For the recipient strain, growth rates were significantly reduced relative to both control groups only in supernatant 6 (mix of *Pediococcus* and *Enterococcus*), with additional significant reductions in other supernatant groups only detected relative to one of the control groups. We therefore find fewer significant effects than in our mixed models analyses, but our key finding that *Pediococcus* spent media reduces growth rates of the plasmid-carrier (donor) relative to control groups still holds for all three *Pediococcus* supernatants.

We have added new text to our Methods section on lines 249-254 and our Results section on lines 287-304, to describe these new analyses and remove all previous

analyses using linear mixed effects models. Full statistical results from the post-hoc pairwise Dunnett tests are found in an additional table: Supplementary Table 1.

- The authors ran their experiments three times (in different 96-well plates) and include this 'batch effect' as a random effect on intercept in their model. However, with only three levels of random effect, it is difficult to estimate the among-block variance (some suggest that random effects require >5 levels, for example Ben Bolker <http://bbolker.github.io/mixedmodels-misc/glmmFAQ.html#should-i-treat-factor-xxx-as-fixed-or-random>). Is there a danger of model overfitting?

The authors should explain how their results are robust to alternative analysis approaches, or explain clearly how their specific approach was planned and justified.

Thank you for your feedback and for highlighting this reference. We had used linear mixed effects models to incorporate both fixed effects and the random effect of 'replicate'. Our random effect was designed to take into account variance in our data distributed across 'blocked' measurements for the set of control and supernatant media for a given replicate, as experiments were performed on different days. Based on simulations from the statistical literature provided in the above link, we now acknowledge that the low number of levels of our random effect (three) is likely to have resulted in overfitting of the data and biased or inaccurate estimates of the random effect variance. We noted that the estimated variance for our random effect was very close to zero for our growth rates mixed effects model. As a result, these models may have increased the likelihood of finding false positives.

To address these limitations, we have performed new statistical analyses without fitting a random effect, for the growth rate and conjugation rate data presented in Figure 1b and c. Growth rate data was analysed using a two-way ANOVA to test the influence of 'Strain' and 'Medium'. Conjugation rate data was analysed using a one-way ANOVA to test the effect of 'Medium'. Following ANOVA tests, we used Dunnett's post-hoc tests to limit pairwise comparisons between individual treatment groups and control groups, for both the donor and recipient strains. See our response to the comment above for the results of the growth rate analyses. We describe all statistical analyses in our Methods section on lines 249- 256.

We find that growth rates significantly varied across 'Medium' groups but now find a non-significant effect of 'Strain', reported in the Results on lines 287-291. For conjugation rates, we did not find a significant effect of 'Medium' or any significant pairwise Dunnett control-treatment group comparisons. The Results section has been adjusted accordingly on lines 339-348 and the sub-heading on lines 337-338 has been adjusted to the following text: "*Metabolic products of lactic acid bacteria do not significantly affect IncI1 plasmid transfer rates*". We have also adjusted the text of our abstract to reflect the non-significant effects on conjugation rates on lines 22-23 and 26 and in the introduction on lines 94-99 as well as in the Discussion on lines 389-390 and 415-417.

The modelling section in the Results has however been largely kept the same, as we illustrated that the effect of conjugation rates on plasmid loss had always been relatively small, as compared to the effect of changes in growth rate. We now argue that – even if the changes in conjugation were real, which in retrospect they do not appear to be – biologically they are not very meaningful. We have edited the Results section on lines 361-367 to reflect this new justification.

Regarding the effect of spent media on growth rates, we find small increases in the growth rate difference between the donor and recipient strains (plasmid fitness cost) in spent media (despite lack of a significant overall strain effect on growth rates in the ANOVA test). We show from the modelling that the magnitude of these effects on growth rate are biologically relevant in determining conditions for plasmid stability, i.e. the increased growth rate difference in spent media is expected to increase chances of plasmid loss. We have edited the Results section on lines 377-383 describing outcomes of the model predictions and in the Discussion on lines 394-395 and 435-437.

Ideally the key strains would also be isolated from the mixed cultures and their effects on plasmid dynamics investigated, but this would be a considerable amount of work.

We agree with the reviewer, that testing the influence of microbiome members on the conjugation process using spent media should be repeated with different donor and recipient strains, including strains isolated from the same microbiome samples. However, our reasoning for using isogenic lab strains as donor and recipient was that there would be a minimal background difference in growth rate. Because differences in growth rate between donor and recipient strain have a major impact on the spread of the plasmid, and since we had no a priori reason to assume growth rate benefits for donor (without plasmid) or recipient-strain background, using strains with the same genetic background would best allow to generalise our findings.

Minor suggestions/comments:

- A clearer justification of why LAB were selected for this study would have been helpful. Although their relative abundance in the chicken caecum environment are stated, putting this into broader context would be useful, such as discussion of other dominant members and their abundances, along with brief information on what is currently known about the microbiome of chicken gut.

Thank you for your comment. We have updated the introduction (lines 80-93) to include further details on the composition of the chicken gut microbiome, to contextualise and highlight the role of lactic acid bacteria in chicken gut health.

- Why were these experiments done under aerobic conditions, when the

majority of gut bacteria are anaerobes? Or does adding the oil to the plate in the plate reader result in an anaerobic culture?

Thank you for your question. Due to practicalities of performing the microtiter plate reader assays (growth rate and conjugation rate assays) under aerobic conditions, we also cultured the gut microbes from the caecal samples under aerobic conditions for consistency. Caecal samples were cultured under aerobic conditions but with sealed tube caps and the oil added to the surface of the cultures in the microtiter plate helped reduce condensation on the plate lid and evaporation (line 190). These two measures would have slightly reduced but not eliminated oxygen supply to the cultures. Our aim was not to exactly replicate conditions of the chicken gut environment in our experiments (which would be difficult to achieve under laboratory conditions). Instead, we contributed conditions of greater biological relevance to plasmid fitness and transfer rate assays that are usually performed in a standard rich laboratory medium, through use of spent media of gut-cultured microbes. This serves as a model for future studies to build upon to investigate the influence of additional factors, including anaerobic conditions, on these key parameters of plasmid stability.

Appendix B

Responses to reviewer comments: manuscript RSPB-2021-2027

Associate Editor

Board Member

Comments to Author:

Thank you for revising your manuscript. Both of the original reviewers are satisfied with the changes you have made. There are a couple of minor corrections/clarifications to make.

We thank the editor and the reviewers for their time and positive feedback. We have addressed the minor corrections, as detailed below.

Reviewer(s)' Comments to Author:

Referee: 1

Comments to the Author(s).

The authors have successfully addressed all my comments.

Thank you for your feedback.

Minor comments:

-line 81, there is an extra "("

This was a typo and we have removed this extra bracket.

-line 86, not sure I get the meaning (is it 10%?).

Yes, this was a typo. We have corrected the text to read '10%'.

Referee: 2

Comments to the Author(s).

In their revision, the authors have added in new data showing the effects on the original (chicken-derived) plasmid-bearing strain, and have reanalysed their results and adjusted their conclusions as a consequence. The changes to the manuscript do lessen the impact slightly, but increase the reliability and generalisability. It is a bit disappointing that (as implied by the conclusion) further studies with increased replication are required to address some of the questions the authors set out to investigate. Nevertheless, I'm satisfied with the changes the authors have made.

Thank you for your feedback.